# The surprising role of the default mode network in naturalistic perception

Talia Brandman 🔾 [1✉], Rafael Malach 🔾 [1] & Erez Simony[1,2]

The default mode network (DMN) is a group of high-order brain regions recently implicated in processing external naturalistic events, yet it remains unclear what cognitive function it serves. Here we identified the cognitive states predictive of DMN fMRI coactivation. Particularly, we developed a state-fluctuation pattern analysis, matching network coactivations across a short movie with retrospective behavioral sampling of movie events. Network coactivation was selectively correlated with the state of surprise across movie events, compared to all other cognitive states (e.g. emotion, vividness). The effect was exhibited in the DMN, but not dorsal attention or visual networks. Furthermore, surprise was found to mediate DMN coactivations with hippocampus and nucleus accumbens. These unexpected findings point to the DMN as a major hub in high-level prediction-error representations.

---

[1] Department of Neurobiology and Azrieli National Institute for Human Brain Imaging and Research, Weizmann Institute of Science, Rehovot 76100, Israel. [2] Faculty of Engineering, Holon Institute of Technology, Holon 5810201, Israel. ✉email: talli.brandman@gmail.com

The default mode network (DMN) is a group of high-order brain regions, so-called for its decreased activation during tasks of high attentional demand, relative to the high baseline activation of the DMN at rest[1–3]. Much research has been conducted in the pursuit of the enigmatic role of this network, consistently pointing to DMN activity during internal processes such as mind wondering, mental time travel, and perspective shifting[4–6]. However, recent neuroimaging studies suggest that the DMN is important not only for internally-driven processes, but remarkably, for long-time scale naturalistic processing of real-life events[7–11], making it central to understanding how our brain tackles incoming information during everyday life. This discovery was enabled by computational advancements in the analysis of neuroimaging signals, which now allow us to track the dynamics of continuous naturalistic processing in healthy human brains, noninvasively[9,12]. Such studies have shown that dynamic responses of the DMN carry information about long timescales of narrative content, and may be associated with subsequent memory of it[7–10]. Yet it remains unknown what are the specific roles of the DMN in naturalistic cognition.

The difficulty in pinpointing the cognitive processes reflected by DMN responses during naturalistic stimulation, lies in connecting between the dynamic cognitive state and DMN activity. Here, we developed a new approach of state-fluctuation pattern analysis (SFPA) to directly relate the two. Specifically, we modeled the cognitive state along the time-course of a movie stimulus using a technique we term retrospective behavioral sampling (see "Materials and methods" section), and compared each cognitive measure to the temporal patterns of neural responses evoked by the same movie. Critically, we employed our previous discovery that task-driven DMN coactivation can be revealed by employing inter-subject functional correlation (ISFC)[9].

Using SFPA to systematically link ISFC to behavior, we were able to show that the cognitive measure that best fits DMN coactivation dynamics is the level of surprise induced by movie events. We further demonstrate surprise-dependent DMN coactivation with subcortical regions implicated in predictive processing[13–16]. This study therefore highlights a surprising role of the DMN, as a central hub in prediction-error representation of ongoing real-life events, likely involving the temporal integration of incoming information with representations stored in memory.

## Results

Cognitive dynamics were modeled from behavioral responses of 45 participants to the first episode of Sherlock (BBC series, 2010), sampling 49 events of the movie on measures of surprise, vividness of memory, emotional intensity and valence, perceived importance, episodic memory, and theory of mind (Fig. 1a, b). Neural dynamics of coactivation (i.e., activity correlations across brain regions), were modeled from functional magnetic resonance imaging (fMRI) responses of 35 participants to the same movie[8,10], in regions of the DMN and hippocampus, as well as the dorsal attention network (DAN) and visual-processing areas (Vis). Since the DMN manifests spontaneous fluctuations both at rest and at task[17–19], we used ISFC to eliminate these spontaneous signals and extract the shared component of stimulus-induced coactivation across brain regions and across individuals[9] (Fig. 1c). Our approach was thus optimized for matching across temporal response patterns of brain and behavior to a dynamic naturalistic input (Fig. 1d).

**Correlations between neural coactivations and the cognitive state across time.** DMN coactivation, both within cortical DMN regions and between DMN and hippocampus, fluctuated proportionally to the magnitude of surprise, but not other behavioral

measures. Particularly, SFPA revealed significant correlations (via permutation test; $p < 0.05$, corrected) between surprise ratings and ISFC among DMN region pairs (Fig. 2a, b). The overall correlation between surprise ratings and ISFC mean across all DMN region pairs was $r(47) = 0.44$ ($p = 0.001$, 95% CI [0.18, 0.64]). In addition, surprise ratings were significantly correlated with ISFC between DMN regions and hippocampus (perm. $p < 0.05$, corrected). By contrast, surprise ratings did not correlate with pairwise ISFC in DAN and Vis (perm. $p > 0.05$, corrected). The overall correlation between surprise ratings and ISFC mean across all regions in DAN was $r(47) = 0.03$ ($p = 0.859$, 95% CI [−0.25, 0.31]) and in Vis was $r(47) = −0.08$ ($p = 0.589$, 95% CI [−0.35, 0.21]). Furthermore, no significant correlations were found between ISFC in the DMN and other behavioral measures (perm. $p > 0.05$, corrected; Supplementary Fig. 1), and effects of surprise were preserved when controlling for behavioral collinearities (Supplementary Fig. 2). Notably, a similar pattern of correlations between ISFC and surprise was also found in participants who had watched a short thriller movie (perm. $p < 0.05$; Supplementary Fig. 3), yet in the thriller, surprise was confounded with emotional intensity ($r(34) = 0.92$, $p < 0.001$, 95% CI [0.85, 0.96]), thus making it uninformative for continued analysis (see Supplementary Note 1).

**Neural coactivations around peak cognitive states.** Independent of the correlation SFPA, peak SFPA examined the relationship between DMN coactivation and peak cognitive states, in an event-triggered analysis. To this end, we extracted the mean network ISFC over time-windows corresponding to the five highest behaviorally-scored events in the movie, separately for each behavioral measure. DMN coactivation was selectively enhanced during peak surprising events (Fig. 3a, b). Particularly, mean ISFC during surprise peaks was higher than during other cognitive peaks in the DMN (perm. $p < 0.05$), but not DAN or Vis (perm. $p > 0.05$). The three networks were significantly different in their ISFC as measured by peak SFPA ($F(12) = 43.94$, $p < 0.001$, $\eta_p^2 = 0.56$; see full ANOVA report in "Methods" section), and particularly during peak surprise ($F(2) = 98.94$, $p < 0.001$, $\eta_p^2 = 0.33$).

Neuro-computational theories of predictive processing describe the brain as a Bayesian inference machine, which optimizes its predictions of future events by calculating the mismatch between expectation and reality, termed prediction error[20,21]. If surprising events in the movie triggered a prediction error, exhibited by increased DMN coactivation, then we would expect the prediction error to decrease with repetition of an initially surprising event. Indeed, we see an example for this in the movie, in a scene depicting a press conference, in which journalists ask police detectives about a series of alleged suicides. At three different times within this press-conference scene, a similar momentary event occurs. The first time, peak surprise is triggered when all journalists at once receive a text message saying "wrong". Later during the scene, the same mass-text event repeats twice more. Each of the three mass-text events corresponded to a separate ISFC window, with no overlap between them, thus we can examine them separately. As demonstrated in Fig. 3c, mean ISFC of the DMN plummeted during the second occurrence of this event, and remained low during the third, whereas DAN and Vis exhibited different response patterns.

**Correlations between surprise ratings and cortical–subcortical coactivations across time.** To further understand the link to predictive processing in our data, we specifically examined striatal regions, which have been previously shown to respond to unexpected stimuli during trial-by-trial learning tasks, and towards

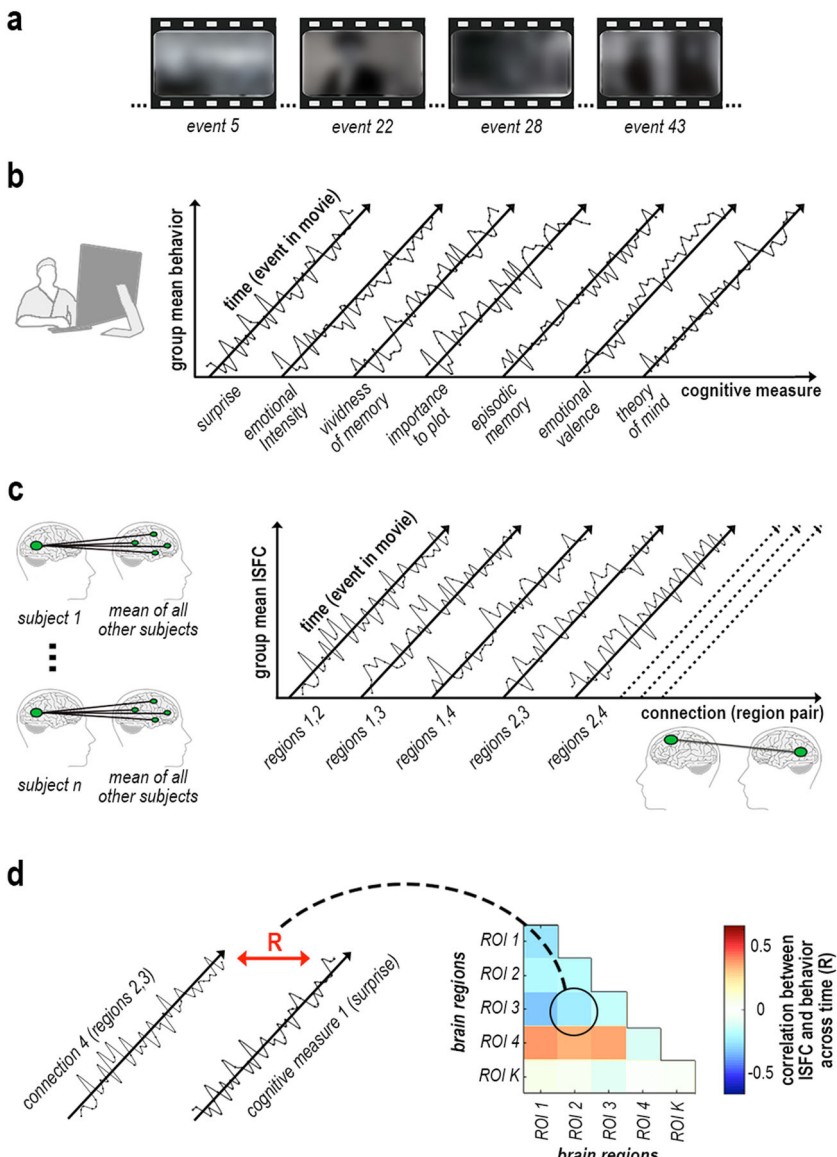

**Fig. 1 State-fluctuation pattern analysis (SFPA) of naturalistic narrative processing.** Illustration of methodology, presenting hypothetical sample stimuli and data. **a** A 23-min-long excerpt of Sherlock (BBC, 2010) was viewed during either fMRI or web-based participation. Retrospective behavioral sampling was performed on 49 movie events. **b** Following web-based viewing, each event was probed verbally in a questionnaire, retrospectively measuring the fluctuation in seven cognitive states throughout the movie. **c** ISFC corresponding to each movie event ±7 TR was measured by correlating the time-window fMRI signal between every region in each participant and every other region in all other participants, measuring the fluctuation in coactivation throughout the movie. **d** Temporal patterns of cognitive states and ISFC were tested for correlation across movie events.

novel contexts in naturalistic stimulation[13–16]. Indeed, SFPA revealed that DMN coactivation with striatal regions, primarily the nucleus accumbens (NAcc), fluctuated proportionally to the state of surprise, as revealed by a significant correlation between surprise and ISFC of NAcc and DMN regions (perm. $p < 0.05$, corrected; Fig. 4a, b). Despite this, surprise did not modulate coactivation among striatal regions themselves, nor between striatum and hippocampus (perm. $p > 0.05$, corrected), consistent with these regions' involvement in a wider range of learning and memory functions[22]. This result did not extend to nearby thalamus, suggesting that surprise-dependent coactivation with DMN is unique to hippocampus and striatum.

**Additional notes and controls.** Notably, the current results cannot be explained by overall DMN activation or deactivation

during surprising events, as no significant correlations were found between surprise ratings and mean univariate responses of DMN regions (perm. $p > 0.05$, corrected), nor did we find univariate effects during surprise peaks (perm. $p > 0.05$). Whole-brain analysis of surprise-dependent univariate activity ($p < 0.05$, corrected) similarly revealed little to no overlap with DMN voxels (Supplementary Fig. 4). Predictive processing is thus reflected in DMN coactivations, i.e., shared patterns of activity fluctuations reflected in ISFC, rather than in a DMN on/off response. Furthermore, low-level stimulus features did not modulate DMN coactivation, as revealed by correlation of visual saliency and luminance with ISFC in DAN and Vis (perm. $p < 0.05$, corrected), but not DMN (perm. $p > 0.05$, corrected; Supplementary Fig. 5).

In addition, we distinguish coactivation magnitude, as ISFC, from its association with behavior, as SFPA. Particularly, low

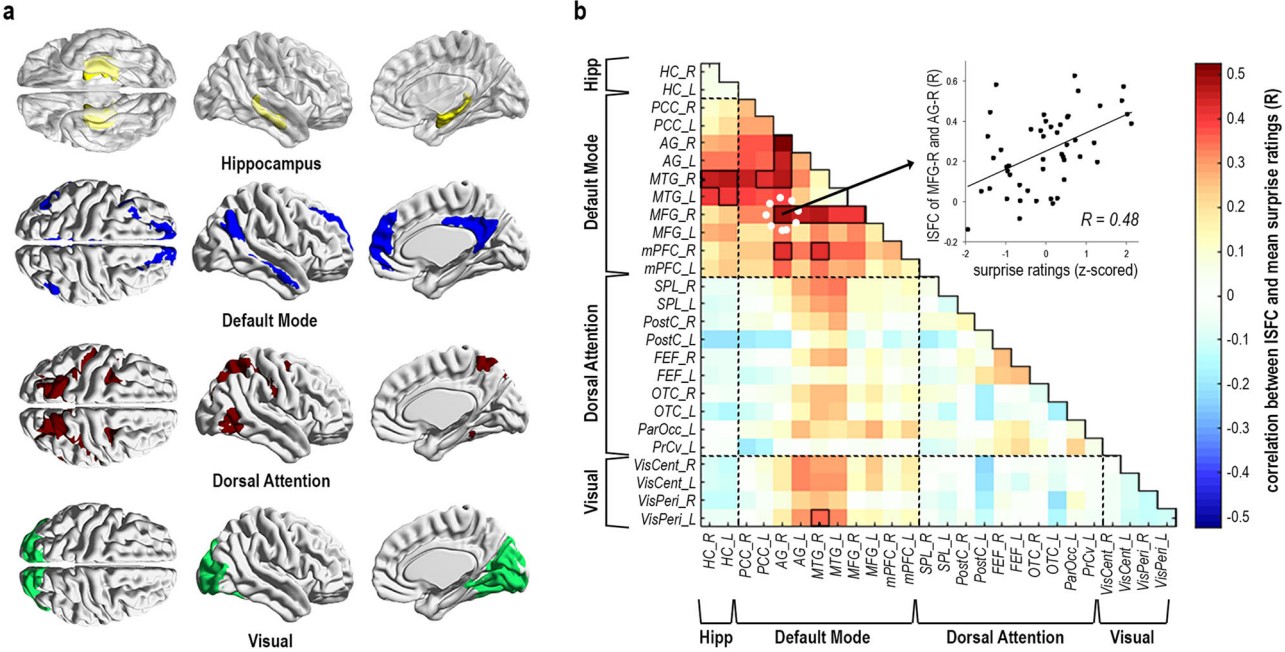

**Fig. 2 Correlation between surprise ratings and coactivations in Sherlock.** Rather than extracting correlations across all brain regions, the analysis was hypothesis-driven, focused on the network of interest (DMN), hippocampus, and two distinct control networks (DAN, Vis), thereby limiting in advance the number of tested comparisons. **a** Brain-maps denoting hippocampus (yellow), DMN (blue), DAN (red), and Vis (green). **b** Correlation SFPA—Pearson correlations were calculated between surprise ratings (mean of 45 behavioral participants) and ISFC of each region-pair (mean of 35 fMRI participants), across the time-course of $n = 49$ movie events. Black outlines denote above-chance correlations at $p < 0.05$ (corrected), determined by random permutation testing (1000 iterations). Scatterplot illustrates the correlation between mean surprise and mean ISFC of the right middle frontal gyrus (MFG) and angular gyrus (AG) across movie events. Regions of interest: HC hippocampus, PCC posterior cingulate cortex, AG angular gyrus, MTG middle temporal gyrus, MFG middle frontal gyrus, mPFC medial prefrontal cortex, SPL superior parietal lobe, PostC postcentral gyrus, FEF frontal eye field, OTC occipital temporal cortex, ParOcc parietal occipital cortex, PrCv precentral ventral region, VisCent visual central areas, and VisPeri visual peripheral areas.

SFPA does not suggest low coactivation in general, as demonstrated by above-chance ISFC (Supplementary Fig. 6) even in region pairs showing below-chance correlation between surprise and ISFC (Fig. 2b). This distinction is particularly evident in the left and right middle temporal gyrus (MTG), showing high ISFC among themselves during peak surprise, though not significantly selective for surprising events (Supplementary Fig. 7a). In fact, surprise explains nearly no variance in these connections, whereas it explains up to ~27% of the variance in ISFC fluctuation of other nodes such as the right angular gyrus (AG) (Supplementary Fig. 7b).

Despite variability across region pairs in explained variance (Supplementary Fig. 7b), note that all DMN regions were nevertheless engaged in surprise-dependent coactivations. Specifically, while AG, MTG (in its subcortical connections) and the middle frontal gyrus (MFG) are the most prominent nodes associated with surprise, also their coactivations with the medial prefrontal cortex (mPFC) and posterior cingulate cortex (PCC) are above chance (Fig. 2b). Selective coactivation of the DMN during peak surprise is also a network-wide effect, preserved after averaging across all network regions (Fig. 3a, b), as well as throughout 49 out of 55 region pairs when examined separately (Supplementary Fig. 7a).

## Discussion
Results reveal coactivations of DMN regions, hippocampus and NAcc, which fluctuate as a function of surprise during naturalistic movie viewing. DMN was further shown to be selectively coactivated during peak surprise, in contrast to other cognitive states. This was found exclusively in DMN, as compared with DAN and Vis, suggesting that surprise ratings are unlikely to reflect low-

level attentional or perceptual processing typical to DAN and Vis[23]. Moreover, because DMN coactivation was not associated with physical stimulus features such as visual luminance and saliency, it is unlikely to reflect low-level sensory processing. Altogether, this suggests that surprise-dependent DMN coactivation reflects a selective high-order response to an unexpected occurrence, as interpreted via semantic processing of movie-narrative content.

The current results are highly compatible with predictive-error related processing[20,21]. This was initially indicated by the reduction in DMN coactivations upon repetition of a previously-surprising event. It suggests that after processing an unusual event for the first time, the prediction error reflected in DMN coactivation is diminished, consistent with error-driven prediction updating[20,21,24]. Furthermore, the engagement of NAcc and hippocampus in surprise-dependent coactivations corresponds with their known roles in error signaling and learning, as part of the dopaminergic reward circuitry[13–15,25]. Thus, surprising movie events appear to be linked to high-level prediction errors, related to semantic processing of the movie narrative. The DMN is central to this process, acting as a hub for surprise-dependent responses of subcortical regions.

To better understand these functional interactions, we consider the role of surprise in semantic comprehension of unfolding events. A surprising event forces us to update our internal model of reality, or in the case of a fictional movie—the narrative, to fit contradictive incoming information. This requires, first, an internal model, second, detection of a mismatch between the internal model and incoming information, and third, integration of incoming information with previously acquired information to improve model predictions. For the first prerequisite, the DMN is

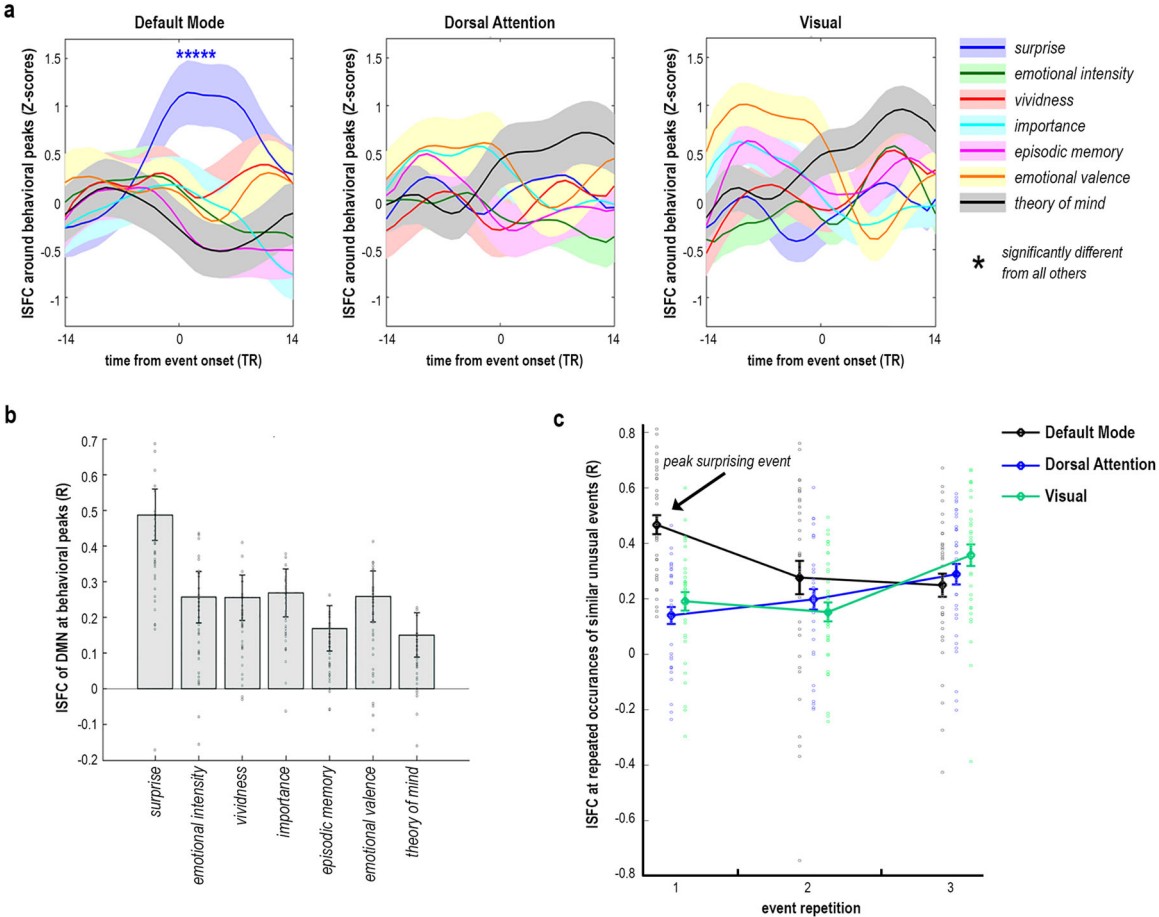

**Fig. 3 Peak analysis of coactivations in Sherlock.** Peak SFPA—mean ISFC time-course of $n = 35$ fMRI participants and of all network regions was averaged across the five peak events on each behavioral measure (e.g., ISFC during five most surprising events). This resulted in a mean ISFC value per network per peak-state, presented here. DMN regions were selectively coactivated during peak surprise, compared to all other peak states, as revealed by random permutation testing (1000 iterations) at $p < 0.05$. Network ISFC is plotted as mean ± SEM across subjects. **a** Peak-SFPA; $t = 0$ corresponds to event onset. Mean ISFC at $t = 0$ was calculated across a 15-TR window centered around t = 0, from −7TR to +7TR. Mean ISFC at $t = 1$ was calculated across a 15-TR window centered around $t = 1$, from −6TR to +8TR, and so on for each plotted time-bin. **b** Peak-SFPA, as mean DMN ISFC corresponding to event onset ($t = 0$ in A). Surprise 0.49 ± 0.07, emotional intensity 0.26 ± 0.07, vividness 0.25 ± 0.06, importance 0.27 ± 0.07, episodic memory 0.17 ± 0.06, emotional valence 0.26 ± 0.07, theory of mind 0.15 ± 0.06. Scattered dots denote individual-subject ISFC values. **c** Mean ISFC across participants and network regions, corresponding to the onsets of three similar events within the same scene, the first of which was found to generate peak surprise. The scene depicted a press conference in which the same initially-surprising text-message was sent to all attendees three times, corresponding to a decrease in mean ISFC of the DMN, but not DAN or Vis. Scattered circles denote individual-subject ISFC values.

a suitable candidate to carry an internal model of the narrative, as its regions have been shown to carry information about narrative content[7–10], and have been hypothesized to represent event models and contextual schemas[4,26,27]. Our findings offer evidence in support of the second prerequisite, i.e., mismatch detection, by showing surprise-dependent coactivation of DMN regions and NAcc, as discussed above. The third step towards model updating, i.e., integration across concurrent and past events, requires the process of memory retrieval. Previous findings have linked DMN regions, as well as their coactivation with hippocampus, to memory recall[10,28,29]. Thus, surprise-dependent coactivation of DMN and hippocampus found here may relate to retrieval processes needed for temporal narrative integration.

This proposed interpretation corresponds with the hypothesis that switching between internal and external based processing modes is necessary for error-driven learning, and involves the DMN and hippocampus[30]. In this case, surprise may lead to switching between unexpected incoming information (external mode), memory of previous events and our internal model (internal modes), as we integrate across all 3. A similar switching

role has been recently hypothesized particularly for the DMN node corresponding to the right AG[31], which is most prominent in its link to surprise in the current study. Altogether, this suggests that coactivation of DMN and subcortical regions support integration across external information and internal representations, as we experience the mental state of surprise.

Finally, our findings are relevant to proposed roles of the DMN that involve prediction and learning, such as social-inference and change-detection[32,33]. By such accounts, the DMN is suggested to monitor the external environment with respect to internal predictions, towards the goal of guiding behavior. Particularly relevant to DMN–NAcc coactivation, reports of functional and neuroanatomical connectivity between NAcc and midline DMN regions[34,35] have been recently integrated into a reinforcement-learning account of DMN functionality, ascribing it explicit stimulus evaluation and prediction roles[31]. By revealing the dynamic connection between the DMN and NAcc as a function of surprise, the current findings are compatible with the coupling of prediction optimization in DMN with the dopaminergic reward circuitry.

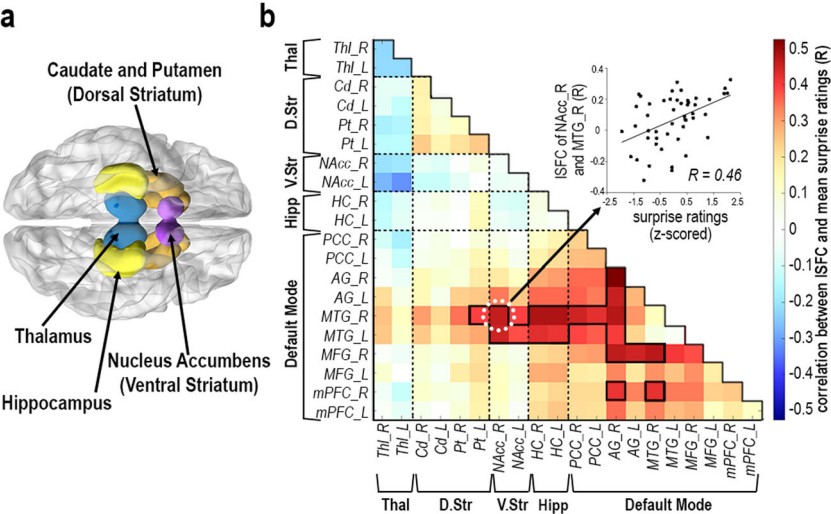

**Fig. 4 Correlation between surprise ratings and cortical-subcortical coactivations in Sherlock. a** Brain-map denoting hippocampus (HC; yellow), nucleus accumbens (NAcc; ventral striatum: purple) caudate and putamen (Cd and Pt; dorsal striatum: Orange), thalamus (Thl; blue). **b** Correlation SFPA—Pearson correlations were calculated between surprise ratings (mean of 45 behavioral participants) and ISFC of each region-pair (mean of 35 fMRI participants), across the time-course of $n = 49$ movie events. Black outlines denote above-chance correlations at $p < 0.05$ (corrected), determined by random permutation testing (1000 iterations). Scatterplot illustrates the correlation between mean surprise and mean ISFC of the right middle temporal gyrus (MTG) and NAcc across movie events. Default mode network regions of interest: PCC posterior cingulate cortex, AG angular gyrus, MTG middle temporal gyrus, MFG middle frontal gyrus, and mPFC medial prefrontal cortex.

## Methods

**Stimuli**. We examined human behavioral responses and functional magnetic resonance imaging (fMRI) responses to two movies. The first movie was a 23-min excerpt[10] from the first episode of the BBC television series Sherlock (2010). The second movie was an 8-min edited excerpt[36,37] from Bang! You're Dead, from the television series Alfred Hitchcock Presents (1961).

**Experimental groups**. The study consisted of four independent experimental groups, one fMRI and one behavioral for each movie, with no known overlap between them: 1. Sherlock fMRI data were obtained with permission from two studies of 17 participants[10] and 18 participants[8], collapsed into a single group of 35 participants; 2. Sherlock behavioral data were obtained from 45 web-based participants; 3. Bang! You're Dead fMRI data were obtained from 30 participants from an open-source repository[36,37]; 4. Bang! You're Dead behavioral data were obtained from 42 web-based participants.

**Behavioral participants**. Forty-five participants (19 female, age $33.2 \pm 8.7$ years) were included in the behavioral data for the movie Sherlock, and 41 participants (17 female, age $31.3 \pm 7.7$ years) were included in the behavioral data for the movie Bang! You're Dead. All participants reported normal or corrected-to-normal vision and hearing, and gave informed consent. Two additional participants for Sherlock, and three additional participants for Bang! You're Dead, were excluded from behavioral-data analysis because they did not complete the task as instructed.

**Behavioral experimental procedure**. We collected behavioral responses to each of the movies using Amazon Mechanical Turk. Experimental procedures were approved by the institutional review board (IRB; approval reference # 533-2) of the Weizmann Institute of Science.

Participants were first screened for technical compatibility (e.g., operating system, internet connection, screen size, and sound) and fluent English writing ability, in order to enable successful video viewing and questionnaire completion. In addition, participants ability to properly hear and see the video was tested before beginning the experiment, in a short audiovisual clip followed by auditory and visual catch questions. Participants were instructed to sit at a distance of 1 foot (12 inches) from the screen. Sherlock was presented at 200 mm over 112.5 mm, and Bang! You're Dead was presented at 180 mm over 135 mm. Participants viewed the movie from start to end without pausing, skipping, or rewinding. Single continuous viewing was additionally monitored via recorded viewing times.

We developed a method of retrospective behavioral sampling in order to measure the fluctuations in cognitive states throughout the movie experience. After viewing the movie, participants first typed a brief free recall describing the content of the movie. Next, participants completed a questionnaire recording their self-reported experience referring to various events of the movie. The questionnaire for Sherlock referred to 49 events, sampling the time-course of the movie at intervals of ~30 s. Because Bang! You're Dead is considerably shorter in time, in order to generate a comparable and effective amount of data points for analysis, we more

densely sampled its questionnaire, referring to 39 events at intervals of ~15 s. Participants were randomly assigned to respond to 1 of 3 subsets of events, chronologically interleaved. Included events were probed in random order throughout the questionnaire. The reminder for each event was presented as a timestamp with a short description of something that happened at a particular moment in the movie (e.g., 10:14 - Sherlock (in lab): "Mike, can I borrow your phone?"). Participants were then asked to focus their memory on that particular event, including no more than a few seconds before and after it. They rated how vividly they remembered the event, typed a detailed free recall of the event, and rated to what extent the event was surprising, emotionally intense, emotionally negative or positive, and important to the plot (Supplementary Table 1). All ratings were collected on scales from 1 to 7. Instructions for the free recall of each event resembled the autobiographical interview method[38,39], asking participants to recall every detail they remembered about what happened at that moment of the movie, what they saw and heard, their thoughts, emotions and physical sensations while viewing the event.

**Behavioral data processing**. We extracted measures of episodic memory and theory of mind (TOM) from the open answers of the free recall for each event separately, as follows. Episodic memory was measured as the number of mentions (memory units) of remembered facts about what had happened in the movie during or adjacent to the event. This score excluded the information already given in the reminder for the event, as well as facts that did not match the actual movie content. In addition, TOM was (orthogonally) measured as the number of references to the state of mind of movie characters during said event. For example, for the event reminder "14:20—Sherlock and Watson enter the flat for the first time", a free recall of the event might be "Sherlock was wearing a blue coat, said he had helped out the landlady, and seemed very proud of himself. Then they stepped into the building". In this case, we mark four stated facts: that Sherlock wore a coat, that it was blue, what he had said about helping the landlady, and that they had entered the building. The last fact was included in the reminder, and the color of the coat does not match the event (it was black), thus both would be excluded, and episodic memory units would be counted as 2. In addition, that Sherlock seemed proud of himself would be counted as 1 TOM unit.

Episodic memory units (integers ≥ 0), TOM units (integers ≥ 0) and each of the behavioral ratings (integers between 1–7) were z-scored (demeaned and divided by standard deviation), within each participant and each behavioral measure separately, across the time-course of responses. Thereafter, responses were averaged across participants, resulting in a single time-course of mean z-scores per behavioral measure, describing the group fluctuation in each cognitive state throughout movie events.

**fMRI data sources**. fMRI data for Sherlock included 17 participants obtained with permission from Chen et al.[10] and 18 participants obtained with permission from Zadbood et al.[8]. These data consisted of preprocessed 3-T fMRI T2*-weighted echo-planar imaging (EPI) blood-oxygen-level-dependent (BOLD) responses with

whole-brain coverage (TR 1500 ms), in Montreal Neurological Institute (MNI) standard volume space[8,10]. Both datasets included the responses to the target sti-mulus, i.e., the first half of the episode, consisting of 946 volumes. Additional data[10] contained responses to the second half of the episode (used for ROI localization), and consisted of 1030 volumes.

fMRI data collection and sharing for Bang! You're Dead was provided by the Cambridge Center for Ageing and Neuroscience (CamCAN)[36,37]. From the repository data, we randomly sampled 30 participants within an age range of 20–50 years. These data consisted of 3-T fMRI T2*-weighted EPI raw BOLD responses with whole-brain coverage[36]. Movie-scan data consisted of 193 volumes (TR 2470 ms). In addition, resting-state data (used for ROI localization) of the same CamCAN participants consisted of 261 volumes (TR 1970 ms).

**fMRI data processing**. fMRI data were analyzed using MATLAB (MathWorks) with statistical parametric mapping (SPM) for preprocessing, NeuroElf for region-of-interest (ROI) organization and BrainNet for ROI visualization.

Preprocessing was performed on raw signals only (CamCAN data) and included slice-timing correction, spatial realignment, transformation to MNI space (voxel size 3 mm × 3 mm × 3 mm), and spatial smoothing with a 6 mm full-width at half-maximum (FWHM) Gaussian kernel. Thereafter, all data underwent voxel-wise detrending and z-scoring across scan volumes.

**ROI localization**. Functional network selection was performed in two steps (Supplementary Fig. 8), constraining selection first by response correlation within tested participant sample, and second by previous functional network definitions based on vast samples[40].

The first selection step measured the response correlation across time, between a single seed region and every voxel in the brain. This was repeated three times with different seeds, one for each of our networks of interest: the default mode network (DMN) seeded in posterior cingulate cortex (PCC: 0, −53, 26), the dorsal attention network (DAN) seeded in the intraparietal sulcus (IPS: 22, −58, 54), and the visual network (Vis) seeded in the primary visual cortex (V1: 30, −88, 0). These seed coordinates were based on a previous study that systematically compared between widely-applied methods for extracting functional brain networks[41], showing that functional connectivity using these seeds (originally from ref.[42]) yielded comparable networks as those resulting from ICA, as well as from alternative seeds. To further assure the stability of the seed, and since the functional regions represented by this seed are typically much larger, we generated a sphere of 80 voxels around the seed coordinates, and used their average signal as seed to measure response correlation. Pearson coefficients were calculated along the time-course (all TRs) of the non-target movie or resting-state scan, separately for each participant, then averaged across all participants. Voxels with mean correlation values of at least 0.3 were selected to continue to the second selection step. In order to maintain comparable network sizes across datasets (DMN, 4418-4768 voxels; DAN 3705-3541 voxels; Vis 4604-4983 voxels), and due to shorter duration (less degrees of freedom) a higher cutoff value of 0.35 was used for CamCAN data.

The second selection step utilized a predefined parcellation of 17 functional networks[40]. The voxels selected for each network in the first selection step were mapped onto the predefined parcellation. Voxels falling outside the network, according to the predefined parcellation, were discarded from network selection. The remaining voxels to pass both the first and second selection steps were included in the final network definition. Finally, ROIs within each network were defined by mapping the selected network voxels to gross anatomical regions according to the predefined network parcellation. Voxels of the DMN were allocated to the PCC/precuneus, angular gyrus (AG), middle temporal gyrus (MTG), middle frontal gyrus (MFG), and medial prefrontal cortex (mPFC). Voxels of the DAN were allocated to the superior parietal lobe (SPL), postcentral gyrus (PostC), frontal eye field (FEF), occipital temporal cortex (OTC), parietal occipital cortex, and precentral ventral region (PrCv). Voxels of Vis were allocated to visual central areas (VisCent) and visual peripheral areas (VisPeri). Independently of functional networks, subcortical ROIs hippocampus (HC), nucleus accumbens (NAcc), caudate (Cd), putamen (Pt), and thalamus (Thl) were defined anatomically via the automated anatomical labeling atlas (AAL)[43].

**ISFC calculation**. To prepare for ISFC analysis, for each participant we extracted the average across voxels within each ROI, along the response time-course of the target-movie scan. We then calculated the average across all other participants (excluding reference participant) across ROI voxels. ISFC between two ROIs was calculated across a sliding time-window of 15 scanning volumes, as the Pearson correlation between the signal time-course of each participant in the first ROI, and the average time-course of all other participants in the second ROI. This correla-tion was calculated separately for each TR (center of time-window ±7 TR) along the time-course of the target movie scan. Correlation values were Fisher-transformed and averaged across participants. This resulted in a single time-course of mean correlations per ROI pair, describing the fluctuation in coactivation among the two ROIs throughout movie events. When the two ROIs were the same region (corresponding to matrix diagonals in figures), the same process resulted in the intersubject correlation (ISC).

**State-fluctuation pattern analysis (SFPA)**. We developed a method of SFPA to examine how cognitive states are dynamically linked to functional network coac-tivation during continuous naturalistic stimulation. The first component of this method is the technique of retrospective behavioral sampling and modeling par-ticipants' natural experience into temporal patterns of cognitive states. The second component of SFPA tests whether dynamic coactivation among brain regions is predicted by each cognitive state. To this end, we present two complementary analyses, which examine the correlation across temporal patterns of coactivation and behavior, and the coactivation corresponding to peak cognitive states. As these analyses were performed across the means of independent groups, for behavior and for coactivation, the temporal patterns of one modality serve as independent predictors for the other.

For the correlation SFPA, we first downsampled the ISFC time-course to match the behavioral time-course, by selecting the ISFC scores centered on each of the behaviorally-tested events in the movie. Thus, each event was assigned a single ISFC score calculated across the 15-TR time-window centered around the behavioral event onset (event TR ± 7). Very early or late events, with less than seven TRs available for ISFC scoring before and after event onset, were discarded, resulting in 49 events for Sherlock, and 36 events for Bang! You're Dead. For each behavioral measure, we then calculated the Pearson correlation between the ISFC time-course (mean of fMRI participants) and each behavioral time-course (mean of behavioral participants). The data points for this correlation were the behaviorally-probed movie events. Thus, the correlation was calculated across events, between the means of the two independent groups. This resulted in a matrix of correlation coefficients as illustrated in Fig. 1d.

For the peak SFPA, we examined the event-triggered ISFC during peak cognitive states. To this end, we first identified the top five peaks along the temporal patterns of cognitive states, for each behavioral measure separately. ISFC values for each region pair were z-scored across the time-course of the movie. We then averaged the ISFC z-scores across all network ROIs, and across the five peak events, within an event window of 29 time-bins centered around the event onset (event TR ± 14). To clarify, the value assigned to each time-bin in the event window is the ISFC score, as calculated across the 15-TR time-window centered around the time-bin TR (for time-window size comparison see Supplementary Fig. 9). For example, time-bin 16 in the event window corresponds to the event TR + 1, and the ISFC score assigned to this time bin was calculated between event TR −6 and event TR + 8. This resulted in a time-course of mean network ISFC, describing the overall network coactivation corresponding to each type of peak cognitive state. In addition, we measured peak-SFPA separately for each DMN region pair, following the same analysis steps, but without averaging ISFC across network regions.

**Statistics and reproducibility**. For correlation SFPA, permutation testing was performed for each ROI pair separately, by random-shuffling the events composing the ISFC time-course and correlating it with the behavioral time-course, repeated 1000 times. This resulted in a distribution of Pearson coefficients, the mean of which is the null hypothesis, i.e., chance-level correlation between ISFC and behavior. The p value of the original Pearson coefficient (of the intact time-courses) was determined by its percentile within the null distribution of Pearson coefficients (of shuffled time-courses), and deemed significant at $p < 0.05$ (two-tailed). Because this was repeated per ROI pair, p values were corrected for multiple comparisons using the false detection rate (FDR)[44].

For peak SFPA, permutation testing was performed to compare between the ISFC of each cognitive state relative to every other state. This was done by measuring the maximum absolute difference between ISFC mean across five randomly-selected events, and ISFC mean across an additional five randomly-selected events, repeated 1000 times. This resulted in the distribution of ISFC maximal differences between two sets of events, the mean of which is the null hypothesis, i.e., that there is no difference between the two sets at the time-bin of maximal difference. The critical threshold to determine significant difference in ISFC between two sets of peak events (e.g., five peak-surprise events versus five peak-vividness events), was determined at $p < 0.05$ (one-tail), corresponding to the 95th percentile of the null distribution of maximal differences.

Internetwork differences in peak SFPA were tested in a repeated-measures ANOVA ($n = 35$ fMRI participants), with ISFC value at event onset (mean across five peaks for each cognitive state) as the dependent variable, and with network (DMN, DAN, and Vis) and cognitive state (surprise, emotional intensity, vividness, importance, episodic memory, emotional valence, and theory of mind) as within-subject factors. Results revealed a significant main effect of cognitive state ($F(6) = 13.79$, $p < 0.001$), a marginal main effect of network ($F(2) = 2.85$, $p = 0.065$), and a highly significant two-way interaction between network and cognitive state ($F(12) = 43.94$, $p < 0.001$). We thus further tested the internetwork differences in peak SFPA specific to surprise, in a repeated-measures ANOVA, with ISFC value at event onset (mean across five surprise peaks) as the dependent variable, and with network (DMN, DAN, and Vis) as within-subject factor, revealing the effect reported in Results.

In addition, significance of peak-SFPA for each region pair was determined in a permutation test, whereby, in each iteration, we randomly picked five events (to match the five peak-state events) and calculated their mean ISFC, assigning the resulting value to a null distribution separately for each pair of DMN regions. This was repeated 1000 times, resulting in a distribution of 1000 data points per region

pair, representing the null hypothesis, i.e., chance-level ISFC across any five events. The $p$ value for each region pair was determined by the percentile of its peak-SFPA in respect to the null distribution. Significance was determined at $p < 0.05$ (one-tail), following FDR correction across all region pairs.

**Control I: SFPA of univariate activations**. To test whether univariate activation or deactivation may explain our results with ISFC, we also performed the SFPA using the mean ROI BOLD time-course in place of the ISFC time-course. The value assigned to each time-bin was the mean BOLD across participants, in the single TR corresponding to it in time. Notably, similar results were found when assigning to each time-bin the mean of the time window corresponding to the ISFC analysis (15 TR) as well as with a 5-TR window. All other analysis steps were the same as in the main analysis. In addition, we conducted a whole-brain analysis by correlating, for each participant, the voxel-wise BOLD with the behavioral time-course of surprise ratings. Pearson coefficients were Fisher-transformed and voxel-wise (Bonferroni-corrected) $T$-test was performed to test group effect. Significant results were plotted on a brain map of $T$-values, describing the magnitude of correlation between BOLD activation and surprise ratings in each voxel (Supplementary Fig. 4c).

**Control II: SFPA of visual attributes**. We tested whether low-level visual features of the movie stimuli were correlated with network coactivation across the same movie events probed in the behavioral experiment. To this end, we extracted the mean levels of visual luminance and spectral saliency from each movie frame, and calculated the average across all movie frames within every time-window corresponding, in temporal range, to the ISFC time-windows. This yielded two time-courses, describing the fluctuations in visual saliency and visual luminance throughout movie events. We then performed SFPA as in the main analysis, using the luminance and saliency time-courses in place of the behavioral cognitive-state time-courses (Supplementary Fig. 5).

**Control III: ISFC of the full-length time-course**. To view overall coactivation magnitudes, irrespective of behavioral fluctuations, we calculated the ISFC of the entire movie time-course, as the Pearson correlation across all scanning 946 volumes (Supplementary Fig. 6a). Permutation testing (as in ref. [9]) was performed by Fourier transforming each participant's fMRI signal, shuffling the phase terms, symmetrizing them, and transforming back (to get a permuted signal while maintaining the autocorrelation of the original signal). Pearson correlation was calculated between the permuted signal and the mean-of-all-but-current participant. Each permutation resulted in an $R$ value for each pair of examined regions, for each of the 35 participants. The maximum absolute value of these was assigned to the null distribution of the max, iterated 1000 times. The mean of this distribution represented the null hypothesis, i.e., chance-level ISFC. The critical threshold for above-chance ISFC (as absolute value) was determined at $p < 0.05$, corresponding to the 95th percentile of the null distribution of the max. In addition, voxel-wise ISC throughout the whole brain were calculated as the Pearson correlation across all 946 scanning volumes, between each voxel in each participant's brain and the same voxel mean-across-all-other participants (Supplementary Fig. 6b).

**Reporting summary**. Further information on research design is available in the Nature Research Reporting Summary linked to this article.

## Data availability

All behavioral data collected in this study are available from the corresponding author on reasonable request. Inquiries regarding fMRI datasets supporting the current findings should be addressed to the relevant research groups[8,10,36,37]. Applications for data access to the CamCAN repository can be made at https://camcan-archive.mrc-cbu.cam.ac.uk/dataaccess/

## Code availability

All codes used for data analysis, and for the collection of behavioral data, are available from the corresponding author on reasonable request.

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

## Acknowledgements

We thank Avigail Mirsky for data curation contribution, Chen et al.[10] and Zadbood et al.[8] for resource contributions, and Aya Ben-Yakov, Talya Sadeh, Michal Bernstein, and Galit Yovel for their useful advice. This work was supported by the Israel Science Foundation grant 1458/17 to E.S. Notably, CamCAN funding was provided by the UK Biotechnology and Biological Sciences Research Council (grant number BB/H008217/1), together with support from the UK Medical Research Council and University of Cambridge, UK.

## Author contributions

Conceptualization and writing by T.B., R.M., and E.S.; Funding acquisition by E.S.; Supervision by R.M. and E.S.; Methodology, project administration, and code programming by T.B. and E.S.; Data curation, formal analysis, investigation, validation, and visualization by T.B.

## Competing interests

The authors declare no competing interests.
