## [Peer Review File · Communications Biology]

Reviewers' comments:

Reviewer #1 (Remarks to the Author):

Brandman and colleagues revisit movie data such as from a 23-minute-long excerpt of Sherlock (BBC, 2010) was viewed during either fMRI or web-based participation. Retrospective behavioral sampling was performed on 49 movie events. (B) Following web-based viewing, each event was probed verbally in a questionnaire, retrospectively measuring the fluctuation in 7 cognitive states throughout the movie. Forty-five participants are examined to study the specific roles of the DMN in naturalistic cognition. The stimulus material and properties resulted from significant effort as behavioral responses to each of the movies using Amazon Mechanical Turk.

Overall, the project is novel and laudable, and well suited to the broad readership of Communications Biology. Aet there are several, some of which severe, flaws that hamper reaching the full potential of the idea and analysis workflow:

Major

1) Methods: ROI definition was based on 80 voxel seeds and based on previously published coordinates. Please provide detailed justifications for both choices, as every subsequent step in the investigation hinges on these.

More so, the two steps of target region creation are hard to follow. Please simplify and consider providing a workflow figure, in the body of the manuscript or in the supplement.

2) Methods: the core of the pattern analysis is based on Pearson correlation coefficients - please go into more detail what was correlated how in much subjects. For instance, the part on the non-parametric null hypothesis testing should be extended, such as by stating in concrete verbalized form what the empirical null hypothesis reflects that the actual data was tested against. As another example, please provide the necessary critical detail to understand the ANOVA including what exactly was used as the continuous outcome measures and how the cognitive states were encoded to fit the ANOVA analysis. As a last example, this reviewer missed mention of treatment of possible confounding influences such as sex, gender and IQ differences. Broadly, please make explicit across which entity Pearson correlation is applied (e.g. subjects, scans, brain regions, etc.) throughout the entire article.

3) Interpretation / "Network coactivation was selectively correlated with the state of surprise across movie events", "This study therefore highlights a new role of the DMN, as a central hub in prediction-error representation of ongoing real-life events": A strong and robust relation between the nucleus accumbens of the reward circuitry and nodes of the default mode network has been reported before and theoretically integrated with other accounts of putative function of the default mode network. Moreover, a different version of the "surprise" narrative is probably the monitor role of the DMN, which has been emphasized by other groups before. Please incorporate and reconcile these and similar literature:

Dohmatob, E., Dumas, G., Bzdok, D., 2020. Dark Control: The Default Mode Network as a Reinforcement Learning Agent. Hum Brain Mapp, in press.

Krienen, F.M., Tu, P.C., Buckner, R.L., 2010. Clan mentality: evidence that the medial prefrontal cortex responds to close others. J Neurosci 30, 13906-13915.

4) Methods: Please provide a rational why one movie was cut into 30s pieces, while the other was cut into 15s windows.

5) Methods: please provide more detailed information on nature and encoding of the scales extracted from the movies (clips).

6) Methods: 45 subjects are mentioned early on, later 17 + 18 subjects, then later again 30 subjects are mentioned. Please make more clear which subjects were contributed to which part of the study, as well as whether these "group" are partially overlapping or not.

7) Results: please remove substantive interpretation of the findings from the results section as much as possible. These contextualizations in the broader literature should be better placed in the discussion section.

8) Epistemology: throughout, please refrain from suggesting or allowing for mis-interpretation of directionality in the conducted analyses and findings, such as here "Thus, the unique element connecting surprise to hippocampus and NAcc here is the DMN."

Minor

1) Please avoid abbreviations in figure titles, such as 'SFPA', to make them more autonomous.

2) Figure 2 caption: ...should explain in one sentence why not the whole brain was analyzed and is shown here. Additionally, it would be more explicit to say across what entity the functional-connectivity-surprise correlations were computed, such as participants, or scenes, or other?

3) Figure 3 caption: please make clear what functional connectivity in the DMN is actually used to test for correspondence with surprise and the other dimensions like vividness etc.

4) z-scoring is defined more than once in the manuscript. Consider trimming down to definition at first mention.

Reviewer #2 (Remarks to the Author):

In the manuscript "the Surprising Role of the Default Mode Network" Brandman et al. look at stimulus locked information-flow between nodes of the DMN and other brain regions which are modulated by subjects' subjective rating of surprise (prediction errors). The manuscript beautifully demonstrates the centrality of the DMN in signaling prediction error, and map for the first time, how few nodes in the DMN, including the angular gyrus, posterior cingulate and middle frontal gyrus, convey surprise-related information to parts of the hippocampus, dorsal attention network, visual network and ventral striatum. The paper is likely to have a big impact on the community given the growing centrality of RL-models in modeling high-level cognitive tasks. Connecting the DMN to the RL-circuits can explain how the error signal is computed and propagated across brain regions, while we process natural real-life stimuli.

I have, however, few comments and suggestions for how to improve the clarity of the paper:

1. The paper solely focusses on the propagation of the surprise signal across brain areas. However, as the authors acknowledge in the discussion, the DMN is involved in many cognitive processes, some may be needed to compute signal errors. The narrow focus is driven by the authors decision to only present the second order correlations between ISFC and the behavioral ratings.

I encourage the authors to present the first order ISFC correlation matrix and the ISC diagonal whole brain maps. I recommend plotting these graphs before and after the projection of the behavioral assessment of surprise.

I will give few specific examples to help the authors understand my point.

First, let us focus on the diagonal (within region ISC reliability). In Figure 2, the MTG-L has a low second order correlation value. Is this value the result of a low ISC in the MTG-L while watching the movie, or is it simply because its reliable activity does not correlate with the surprise ratings?

Now let's look at the right Angular gyrus (AG_R). It seems to strongly correlate with the surprise ratings. But how much of the variance is explained by the surprise ratings? In other words, if you would project out the surprise ratings would you get zero ISC? Or would that remove only a small portion of the variance?

Second, the same analysis will also be revealing for the connectivity portion of the matrix. For example, both AG_L-HC_L and AG_L-MFG_R strongly correlate with the surprise ratings. But how much of the ISFC is explained by the surprise ratings? Calculating the ISFC between these regions after the projection of the surprise ratings can easily address this question.

Note, even in the case that the surprise ratings explained only small portion of the variance in the inter subject functional correlation, it will not downplay the importance of your findings. At the same time, adding this information can help to contextualize your findings.

2. The paper indicates that not all DMN nodes contribute equally to the propagation of surprise ratings across regions. For example, AG and PCC seem more central than other nodes. Recently few studies subdivided the DMN into subnetworks (e.g. see recent bio archive paper by Barnett and Ranganath). Does the propagation of surprise ratings match one of these subdivisions better?

Minor comments:

1. I was worried that using a narrow window of 15TRs will make the ISFC analysis unstable. Especially in Figure 3. Would it be possible to add a supplementary analysis with window sizes of 30TRs and 60TRs?

2. There appear to be some similarities in the connections between hippocampus and DMN during the ratings of surprise and during the ratings of importance and emotional intensity. Did you control for possible partial correlations across the behavioral ratings?

3. I was confused by the wording you used to describe Fig 3C: "repetitions of a similar event within the same scene". I assume the same event was repeated across different scenes, and not that the same event was repeated three times within the same scene. If it were repeated within the same scene how could you detect the difference between the three repetitions within the same time window?

Dear Communications Biology Editor and Reviewers,

We thank the reviewers for their thorough evaluation and useful suggestions, and have aimed to address their concerns in full. We believe the revised submission to be improved in both comprehensiveness and clarity. See below our detailed reply to each comment.

- Reviewer comments are highlighted in gray, quotes from the manuscript are indented and italicized, and changes to the original text are in red.
- New supplementary items: Supplementary Table 1; Supplementary Figures

2,6,7,8,9. Reviewer #1 (Remarks to the Author):

Major

1) Methods: ROI definition was based on 80 voxel seeds and based on previously published coordinates. Please provide detailed justifications for both choices, as every subsequent step in the investigation hinges on these.

More so, the two steps of target region creation are hard to follow. Please simplify and consider providing a workflow figure, in the body of the manuscript or in the supplement.

The seed coordinates (originally from Andrews-Hanna et al., 2007) were based on previous validation in a systematic comparison (Van Dijk et al., 2010), revealing comparable networks using alternative seeds and using ICA. To further assure the stability of the seed we generated large 80 voxel spheres. Since PCC, IPS and V1 are of typically large cortical volume, we expected the large seeds were still well within these regions' core. In addition, we have now tested the response correlations based on 7-voxel seeds, relative to the original 80-voxel seeds, to make sure results are stable across different seed sizes. We generated the 7-voxel seeds by creating smaller spheres around the same center coordinates. The voxel-wise response correlation map (mean across subjects), with the same cutoff of $R > 0.3$ as in the original analysis (step 1 in the network selection process clarified below), is in almost full overlap when comparing the two seed sizes. Thus, we are assured that seed size was of no major consequence to our reported effects, as even large variations in seed size result in almost identical voxel-wise network definition.

In the manuscript, we have clarified the ROI selection process, as well as the rationale for seed definition in Methods, lines 335-371. Following the reviewer's suggestion, we now also

provide a new Supplementary Figure 8 illustrating the ROI selection process. Both are copied below.

ROI localization

Functional network selection was performed in two steps (Supplementary Figure 8), constraining selection first by response correlation within tested participant sample, and second by previous functional network definitions based on vast samples⁴⁰.

The first selection step measured the response correlation across time, between a single seed region and every voxel in the brain. This was repeated 3 times with different seeds, one for each of our networks of interest: the default mode network (DMN) seeded in posterior cingulate cortex (PCC: 0, -53, 26), the dorsal attention network (DAN) seeded in the intraparietal sulcus (IPS: 22, -58, 54), and the visual network (Vis) seeded in the primary visual cortex (V1: 30, -88, 0). These seed coordinates were based on a previous study that systematically compared between widely-applied methods for extracting functional brain networks⁴¹, showing that functional connectivity using these seeds (originally from⁴²) yielded comparable networks as those resulting from ICA, as well as from alternative seeds. To further assure the stability of the seed, and since the functional regions represented by this seed are typically much larger, we generated a sphere of 80 voxels around the seed coordinates, and used their average signal as seed to measure response correlation. Pearson coefficients were calculated along the time-course (all TRs) of the non-target movie or resting scan, separately for each participant, then averaged across all participants. Voxels with mean correlation values of at least 0.3 were selected to continue to the second selection step. In order to maintain comparable network sizes across datasets (DMN, 4418-4768 voxels, DAN 3705-3541 voxels, Vis 4604-4983 voxels), and due to shorter duration (less degrees of freedom) a higher cutoff value of 0.35 was used for CamCAN data.

The second selection step utilized a predefined parcellation of 17 functional networks⁴⁰. The voxels selected for each network in the first selection step were mapped onto the predefined parcellation. Voxels falling outside the network according to the predefined parcellation were discarded from network selection. The remaining voxels to pass both the first and second selection steps were included in the final network definition. Finally, ROIs within each network were defined by mapping the selected network voxels to gross anatomical regions according to the predefined network parcellation. Voxels of the DMN were allocated to the PCC/precuneus, angular gyrus (AG), middle temporal gyrus (MTG), middle frontal gyrus (MFG), and medial prefrontal cortex (mPFC). Voxels of the DAN were allocated to the superior parietal lobe (SPL), postcentral gyrus (PostC), frontal eye field (FEF), occipital temporal cortex (OTC), parietal occipital cortex, and precentral ventral region (PrCv). Voxels of Vis were allocated to visual central areas (VisCent) and visual peripheral areas (VisPeri). Independently of functional networks, subcortical ROIs hippocampus (HC), nucleus accumbens (NAcc), Caudate (Cd), Putamen (Pt), and thalamus (Thl) were defined anatomically via the automated anatomical labelling atlas (AAL)⁴³.

Supplementary Figure 8. Region of interest (ROI) localization. Workflow, from left to right: 1. Seed for functional localization of the network was defined anatomically as a sphere around MNI coordinates validated in previous reports¹; 2. Whole-brain voxel-wise network definition, first, precluded voxels of low mean response correlation with the seed region (during a non-target scan), and second, precluded voxels mapped outside a network atlas that had been functionally defined based on a wide sample¹. Voxels passing both filtering stages were selected for the final network definition; 3. For ROI definition, network voxels were allocated to general anatomical regions mapped to the predefined network atlas¹. ¹Schaefer, A. et al. Local-Global Parcellation of the Human Cerebral Cortex from Intrinsic Functional Connectivity MRI. *Cereb Cortex* 28, 3095-3114, doi:10.1093/cercor/bhx179 (2018).

Pearson correlations were used in our study in 3 of our analysis steps:

1. During the first step of network ROI definition, seed-region signal was correlated with every voxel within each participant's brain separately, then averaged across fMRI participants. The data points for correlation were all TRs comprising the time-course of the non-target scan (see step 1 of ROI definition clarified in previous comment).

2. For calculating the ISFC of fMRI data, each ROI in a single subject was correlated with the mean-across-all-other-subjects of every ROI. The data points for this correlation were the 15 TRs comprising the sliding time-window. This correlation was performed separately for each TR (as center of time-window) along the time-course of the target movie scan. This ISFC time-course was then averaged across all fMRI participants. This has now been clarified in Methods, lines 376-384:

... ISFC between two ROIs was calculated across a sliding time-window of 15 scanning volumes, as the Pearson correlation between the signal time-course of each participant in the first ROI, and the average time-course of all other participants in

the second ROI. This correlation was calculated separately for each TR (center of time-window ± 7 TR) along the time-course of the target movie scan. Correlation values were Fisher-transformed and averaged across participants. This resulted in a single time-course of mean correlations per ROI pair, describing the fluctuation in coactivation among the two ROIs throughout movie events. When the two ROIs were the same region (corresponding to matrix diagonals in figures), the same process resulted in the inter-subject correlation (ISC).

3. For calculating the SFPA between brain and behavior, the ISFC time-course (mean across all fMRI subjects) was correlated with each behavioral measure (mean across all behavioral subjects). Thus, the correlation is between the *means* of the two independent groups. The data points for this correlation were the behaviorally probed movie events (in Sherlock – 49 events). This has now been clarified in Methods, lines 403-408:

... we then calculated the Pearson correlation between the ISFC time-course (mean of fMRI participants) and each behavioral time-course (mean of behavioral participants). The data points for this correlation were the behaviorally probed movie events. Thus, the correlation was calculated across events, between the means of the two independent groups. This resulted in a matrix of correlation coefficients as illustrated in Figure 1D.

Additionally, we extended, as requested, the descriptions of permutation testing and ANOVA, now presented together under Statistics and Reproducibility in Methods, lines 424453:

For correlation SFPA, permutation testing was performed for each ROI pair separately, by random-shuffling the events composing the ISFC time-course and correlating it with the behavioral time-course, repeated 1000 times. This resulted in a distribution of Pearson coefficients, the mean of which is the null hypothesis, i.e. chance-level correlation between ISFC and behavior. The p value of the original Pearson coefficient (of the intact time-courses) was determined by its percentile within the null distribution of Pearson coefficients (of shuffled time-courses), and deemed significant at $p < 0.05$ (2-tailed). Because this was repeated per ROI pair, p values were corrected for multiple comparisons using the false detection rate (FDR)
44.

For peak SFPA, permutation testing was performed to compare between the ISFC of each cognitive state relative to every other state. This was done by measuring the maximum absolute difference between ISFC mean across 5 randomly-selected events, and ISFC mean across an additional 5 randomly-selected events, repeated 1000 times. This resulted in the distribution of ISFC maximal differences between two sets of events, the mean of which is the null hypothesis, i.e. that there is no difference between the two sets at the time-bin of maximal difference. The critical threshold to determine significant difference in ISFC between two sets of peak events (e.g. 5 peak-surprise events versus 5 peak-vidviness events), was determined at $p < 0.05$ (1-tail), corresponding to the 95th percentile of the null distribution of maximal differences.

Inter-network differences in peak SFPA were tested in a repeated-measures ANOVA, with ISFC value at event onset (mean across 5 peaks for each cognitive state) as the dependent variable, and with network (DMN, DAN, Vis) and cognitive state (surprise, emotional intensity, vividness, importance, episodic memory, emotional valence, theory of mind) as within-subject factors. Results revealed a significant main effect of

cognitive state ($F(6) = 13.79, p < 0.001$), a marginal main effect of network ($F(2) = 2.85, p = 0.065$), and a highly significant two-way interaction between network and cognitive state ($F(12) = 43.94, p < 0.001$). We thus further tested the inter-network differences in peak SFPA specific to surprise, in a repeated-measures ANOVA, with ISFC value at event onset (mean across 5 surprise peaks) as the dependent variable, and with network (DMN, DAN, Vis) as within-subject factor, revealing the effect reported in results.

Finally, we address the reviewer's concern about possible confounding demographic influences. All experimental groups included heterogeneous samples of both men and women across a wide age range (cut off criteria 18-50 years), with no additional demographic constraints. Because SFPA effects depend on the match between the means of two independent groups of subjects recruited in different contexts (fMRI group and web-survey group), the large heterogeneity within and between these groups reduces the risk of confounding demographic influences and increases generalizability. This is because artificially matching demographics across groups would run the risk of artificially increasing the correlations between them, whereas demographic confounds are unlikely to drive high correlations between two demographically heterogeneous samples.

3) Interpretation / "Network coactivation was selectively correlated with the state of surprise across movie events", "This study therefore highlights a new role of the DMN, as a central hub in prediction-error representation of ongoing real-life events": A strong and robust relation between the nucleus accumbens of the reward circuitry and nodes of the default mode network has been reported before and theoretically integrated with other accounts of putative function of the default mode network. Moreover, a different version of the "surprise" narrative is probably the monitor role of the DMN, which has been emphasized by other groups before. Please incorporate and reconcile these and similar literature:

Dohmatob, E., Dumas, G., Bzdok, D., 2020. Dark Control: The Default Mode Network as a Reinforcement Learning Agent. *Hum Brain Mapp*, in press.

Krienen, F.M., Tu, P.C., Buckner, R.L., 2010. Clan mentality: evidence that the medial prefrontal cortex responds to close others. *J Neurosci* 30, 13906-13915.

We completely agree that the newly published Markov decision model of the DMN is highly relevant to the current findings (though we cannot refer to learning, per se), as are other accounts of DMN function that involve switching between internal predictions and external stimuli. We have integrated these into the Discussion, lines 223-232:

Finally, our findings are relevant to proposed roles of the DMN that involve prediction and learning, such as social-inference and change-detection (e.g. Krienen et al., 2010, Pearson et al., 2011). By such accounts, the DMN is suggested to monitor the external environment with respect to internal predictions, towards the goal of guiding behavior. Particularly relevant to DMN-NAcc coactivation, reports of functional and neuroanatomical connectivity between NAcc and midline DMN regions (Croxson et al., 2005, Bzdok et al., 2015) have been recently integrated into a reinforcement-learning account of DMN functionality, ascribing it explicit stimulus evaluation and prediction roles (Dohmatob et al., 2020). By revealing the dynamic connection between the DMN and NAcc as a function of surprise, the current findings

are compatible with the coupling of prediction optimization in DMN with the dopaminergic reward circuitry.

4) Methods: Please provide a rational why one movie was cut into 30s pieces, while the other was cut into 15s windows.

The two movies were considerably different in length (23 min / 8 min), thus to generate a comparable and effective amount of data points for SFPA we more densely sampled the shorter of the two. Now explained in Methods, lines 272-276:

The questionnaire for Sherlock referred to 49 events, sampling the time-course of the movie at intervals of -'30 sec. Because Bang! You're Dead is considerably shorter in time, in order to generate a comparable and effective amount of data points for analysis, we more densely sampled its questionnaire, referring to 39 events at intervals of -'15 seconds.

0)Methods: please provide more detailed information on nature and encoding of the scales extracted from the movies (clips).

We have added a new Supplementary Table 1 (copied below) with the behavioral instructions of the scale ratings and free recall. In addition, we now include more details on the encoding process in Methods, lines 290-310:

We extracted measures of episodic memory and theory of mind (TOM) from the open answers of the free recall for each event separately, as follows. Episodic memory was measured as the number of mentions (memory units) of remembered facts about things that happened in the movie during or adjacent to the event. This score excluded the information already given in the reminder for the event, as well as facts that did not match the actual movie content. In addition, TOM was (orthogonally) measured as the number of references to the state of mind of movie characters during said event. For example, for the event reminder "14:20 - Sherlock and Watson enter the flat for the first time", a free recall of the event might be "Sherlock was wearing a blue coat, said he had helped out the landlady, and seemed very proud of himself. Then they stepped into the building". In this case, we mark 4 stated facts: that Sherlock wore a coat, that it was blue, what he had said about helping the landlady, and that they had entered the building. The last fact was included in the reminder, and the color of the coat does not match the event (it was black), thus both were would be excluded, and Episodic memory units would be counted as 2. In addition, that Sherlock seemed proud of himself would be counted as 1 TOM unit.

Episodic memory units (integers ? 0), TOM units (integers ? 0) and each of the behavioral ratings (integers between 1-7) were z-scored (demeaned and divided by standard deviation), within each participant and each behavioral measure separately, across the time-course of responses. Thereafter, responses were averaged across subjects, resulting in a single time-course of mean z-scores per behavioral measure, describing the group fluctuation in each cognitive state throughout movie events.

Measure	Questionnaire Instructions	Scale min – 1	Scale max – 7
	Please take a moment to recall this moment Vividness of the movie. How vivid is your memory of this event?	cloudy and imageless	clear and vivid as if experienced again
Surprise	How surprising was the event?	did not surprise me at all	no other event in the movie surprised me this much
Free recall	Focus on this particular moment in the movie, including no more than a few seconds before and after the described event. Write down every detail you can remember, including any or all of the following types of information: what happened in the movie, what you saw, what you heard, what were your own thoughts, emotions and/or physical sensations while you were watching that event, etc.	n/a	n/a
Emotional intensity	How emotionally intense was the event while watching it?	no detectable emotion	the most intense event to watch in this movie
Emotional valence	Would you rate this event as emotionally positive or negative ?	strongly negative	strongly positive
Importance	How important was this event to the main story of the movie?	insignificant	more important than any other event in the movie

Supplementary Table 1. Questionnaire phrasing of scale rating and free recall instructions. Instructions were repeated identically with every event reminder. Scale ratings were reported via keys 1 through 7 on the computer keyboard. Free recall was typed into an open-ended response field under a 3-minute time-limit per event reminder.

6) Methods: 45 subjects are mentioned early on, later 17 + 18 subjects, then later again 30 subjects are mentioned. Please make more clear which subjects were contributed to which part of the study, as well as whether these "group" are partially overlapping or not.

We agree that due to having several different data sources, this is confusing. We have therefore added a sub-section in Methods, lines 240-247, clarifying in one place the allocation to experimental groups.

Experimental Groups

The study consisted of 4 independent experimental groups, one fMRI and one behavioral for each movie, with no known overlap between them: 1. Sherlock fMRI data were obtained with permission from two studies of 17 participants¹⁰ and 18

participants⁸, collapsed into a single group of 35 participants; 2. Sherlock behavioral data were obtained from 45 web-based participants; 3. Bang! You're Dead fMRI data were obtained from 30 participants from an open-source repository^{36,37}; 4. Bang! You're Dead behavioral data were obtained from 42 web-based participants.

As before, full description of behavioral participants and fMRI data sources can be found in Methods, lines 248-254, 311-319.

7) Results: please remove substantive interpretation of the findings from the results section as much as possible. These contextualizations in the broader literature should be better placed in the discussion section.

We agree that interpretations would be better placed within the Discussion. We therefore made the following changes, removing parts from Results and expanding in Discussion:

Removed from Results:

...This suggests that surprise ratings are unlikely to reflect a large attentional shift or low-level perceptual processing typical to Vis and DAN (20), and thus more likely to reflect a higher-order response to an unexpected occurrence.

...This suggests that after processing an unusual event for the first time, the prediction error reflected in DMN coactivation is diminished, consistent with error-driven prediction updating (21-23).

...Thus, DMN coactivations are unlikely to reflect low-level sensory processing, and more likely correspond to higher semantic processing of movie-narrative content.

Expanded in Discussion, lines 178-197:

Results reveal coactivations of DMN regions, hippocampus and NAcc, which fluctuate as a function of surprise during naturalistic movie viewing. DMN was further shown to be selectively coactivated during peak surprise, in contrast to other cognitive states. This was found exclusively in DMN, as compared with DAN and Vis, suggesting that surprise ratings are unlikely to reflect low-level attentional or perceptual processing typical to DAN and Vis²³. Moreover, because DMN coactivation was not associated with physical stimulus features such as visual luminance and saliency, it is unlikely to reflect low-level sensory processing. Altogether, this suggests that surprise-dependent DMN coactivation reflects a selective high-order response to an unexpected occurrence, as interpreted via semantic processing of movie-narrative content.

The current results are highly compatible with predictive-error related processing^{20,21}. This was initially indicated by the reduction in DMN coactivations upon repetition of a previously-surprising event. It suggests that after processing an unusual event for the first time, the prediction error reflected in DMN coactivation is diminished, consistent with error-driven prediction updating^{20,21,24}. Furthermore, the engagement of NAcc and hippocampus in surprise-dependent coactivations corresponds with their known roles in error signaling and learning, as part of the dopaminergic reward circuitry^{13-15,25}. Thus, surprising movie events appear to be linked to high-level prediction errors, related to semantic processing of the movie narrative. The DMN is central to this process, acting as a hub for surprise-dependent responses of subcortical regions.

8) Epistemology: throughout, please refrain from suggesting or allowing for mis-interpretation of directionality in the conducted analyses and findings, such as here "Thus, the unique element connecting surprise to hippocampus and NAcc here is the DMN."

Indeed, we do not intend to infer directionality. We have removed the above statement and checked the phrasing of the revised manuscript to avoid potential confusion.

Minor

1) Please avoid abbreviations in figure titles, such as 'SFPA', to make them more autonomous.

We have expanded abbreviations in figure titles.

2) Figure 2 caption: ...should explain in one sentence why not the whole brain was analyzed and is shown here. Additionally, it would be more explicit to say across what entity the functional-connectivity-surprise correlations were computed, such as participants, or scenes, or other?

We have clarified both points in the figure caption:

Figure 2. Correlation *between* surprise ratings and coactivations in Sherlock. *Rather than extracting correlations across all brain regions, the analysis was hypothesis-driven, focused on the network of interest (DMN), hippocampus, and two distinct control networks (DAN, Vis), thereby limiting in advance the number of tested comparisons. Left: brain-maps denoting Hippocampus (yellow), DMN (blue), DAN (red) and Vis (green). Right: Correlation SFPA – Pearson correlations were calculated between surprise ratings (mean of 45 behavioral participants) and ISFC of each region-pair (mean of 35 fMRI participants), across the time-course of 49 movie events. Black outlines denote above-chance correlations at $p < 0.05$ (corrected), determined by random permutation testing (1000 iterations). Scatterplot illustrates the correlation between mean surprise and mean ISFC of the right middle frontal gyrus*

3) Figure 3 caption: please make clear what functional connectivity in the DMN is actually used to test for correspondence with surprise and the other dimensions like vividness etc.

(MFG) and angular gyrus (AG) across movie events.

We have clarified this point in the figure caption:

Figure 3. Peak *analysis* of coactivations in Sherlock. *Peak SFPA – mean ISFC time-course of 35 fMRI participants and of all network regions was averaged across the 5 peak events on each behavioral measure (e.g. ISFC during 5 most surprising events). This resulted in a mean ISFC value per network per peak-state, presented here. DMN regions were selectively coactivated during peak surprise, compared to all other peak states, as revealed by random permutation testing (1000 iterations) at $p < 0.05$. Network ISFC is plotted as mean \pm SEM across subjects. (A) Peak-SFPA; $t=0$ corresponds to event onset. Mean ISFC at $t=0$ was calculated across a 15-TR window centered around $t=0$, from $-7TR$ to $+7TR$. Mean ISFC at $t=1$ was calculated across a*

15-TR window centered around $t=1$, from $-6TR$ to $+8TR$, and so on for each plotted time-bin. (B) Peak-SFPA, as mean DMN ISFC corresponding to event onset ($t=0$ in

A). *Surprise* 0.49 ± 0.07 , *Emotional Intensity* 0.26 ± 0.07 , *Vividness* 0.25 ± 0.06 , *Importance* 0.27 ± 0.07 , *Episodic Memory* 0.17 ± 0.06 , *Emotional Valence* 0.26 ± 0.07 , *Theory of Mind* 0.15 ± 0.06 . Gray circles denote individual-subject ISFC values. (C) *Mean ISFC across participants and network regions, corresponding to the onsets of 3 similar events within the same scene, the first of which was found to generate peak surprise. The scene depicted a press conference in which the same initially-surprising text-message was sent to all attendees 3 times, corresponding to a decrease in mean ISFC of the DMN, but not DAN or Vis.*

4) z-scoring is defined more than once in the manuscript. Consider trimming down to definition at first mention.

Thank you for spotting this. We have removed the redundant instances of z-scoring definition.

Reviewer #2 (Remarks to the Author):

In the manuscript “the Surprising Role of the Default Mode Network” Brandman et al. look at stimulus locked information-flow between nodes of the DMN and other brain regions which are modulated by subjects’ subjective rating of surprise (prediction errors). The manuscript beautifully demonstrates the centrality of the DMN in signaling prediction error, and map for the first time, how few nodes in the DMN, including the angular gyrus, posterior cingulate and middle frontal gyrus, convey surprise-related information to parts of the hippocampus, dorsal attention network, visual network and ventral striatum. The paper is likely to have a big impact on the community given the growing centrality of RL-models in modeling high-level cognitive tasks. Connecting the DMN to the RL-circuits can explain how the error signal is computed and propagated across brain regions, while we process natural real-life stimuli.

I have, however, few comments and suggestions for how to improve the clarity of the paper:

1) The paper solely focusses on the propagation of the surprise signal across brain areas. However, as the authors acknowledge in the discussion, the DMN is involved in many cognitive processes, some may be needed to compute signal errors. The narrow focus is driven by the authors decision to only present the second order correlations between ISFC and the behavioral ratings.

I encourage the authors to present the first order ISFC correlation matrix and the ISC diagonal whole brain maps.

I recommend plotting these graphs before and after the projection of the behavioral assessment of surprise.

I will give few specific examples to help the authors understand my point.

First, let us focus on the diagonal (within region ISC reliability). In Figure 2, the MTG-L has a low second order correlation value. Is this value the result of a low ISC in the MTG-L while watching the movie, or is it simply because its reliable activity does not correlate with the surprise ratings?

Now let's look at the right Angular gyrus (AG_R). It seems to strongly correlate with the surprise ratings. But how much of the variance is explained by the surprise ratings? In other words, if you would project out the surprise ratings would you get zero ISC? Or would that remove only a small portion of the variance?

Second, the same analysis will also be revealing for the connectivity portion of the matrix. For example, both AG_L-HC_L and AG_L-MFG_R strongly correlate with the surprise ratings. But how much of the ISFC is explained by the surprise ratings? Calculating the ISFC between these regions after the projection of the surprise ratings can easily address this question.

Note, even in the case that the surprise ratings explained only small portion of the variance in the inter subject functional correlation, it will not downplay the importance of your findings. At the same time, adding this information can help to contextualize your findings.

This is an important point. We address it by examining the two main questions brought up by this comment:

- (1) What is the relative contribution of surprise to the entire DMN signal, i.e. the variance explained by surprise in each ISC node and ISFC pair?
- (2) Do low correlations with surprise stem from low reliability of the signal to begin with, as posed in the example of the left MTG?

The short answers are that (1) the variance explained by surprise in the DMN was measured at most at ~27% of dynamic ISFC, and that (2) regions with low correlations with surprise may nevertheless exhibit highly reliable signals, which cannot be explained by surprise, as in the left MTG. In the following paragraphs we discuss each of these points in depth, and report additional analyses suggested by the reviewer.

First, as per the reviewer's suggestion, we now plot the ISFC matrix (Panel A; above-chance outlined in black) and ISC whole-brain map (Panel B; DMN ROIs are outlined in black) calculated across the entire time-course of the movie, as a single time window of 946 TRs (new Supplementary Figure 6):

Supplementary Figure 6. Inter-subject correlations across the entire scan time-course for Sherlock. (A) ISFC between every pair of ROIs, calculated as the Pearson correlation across all 946 scanning volumes. Black outlines denote above-chance correlations at $p < 0.05$,

determined by random permutation testing (1000 iterations); (B) Voxel-wise ISC throughout the whole brain, calculated as the Pearson correlation across all 946 scanning volumes. Black outlines denote the ROIs of the DMN. ISFC and ISC values plotted as means of 35 fMRI participants.

This analysis is now described in Methods, lines 484-499:

Control III: ISFC of the full-length time-course

To view overall coactivation magnitudes, irrespective of behavioral fluctuations, we calculated the ISFC of the entire movie time-course, as the Pearson correlation across all scanning 946 volumes (Supplementary Figure 6A). Permutation testing (as in ⁹) was performed by Fourier transforming each participant's fMRI signal, shuffling the phase terms, symmetrizing them, and transforming back (to get a permuted signal while maintaining the autocorrelation of the original signal). Pearson correlation was calculated between the permuted signal and the mean-of-all-but-current participant. Each permutation resulted in an R value for each pair of examined regions, for each of the 35 participants. The maximum absolute value of these was assigned to the null distribution of the max, iterated 1000 times. The mean of this distribution represented the null hypothesis, i.e. chance-level ISFC. The critical threshold for above-chance ISFC (as absolute value) was determined at $p < 0.05$, corresponding to the 95th percentile of the null distribution of the max. In addition, voxel-wise ISC throughout the whole brain were calculated as the Pearson correlation across all 946 scanning volumes, between each voxel in each participant's brain and the same voxel mean-across-all-other participants (Supplementary Figure 6B).

The ISC/ISFC matrix shown in Panel A is already informative to the second question, demonstrating that all DMN regions present highly reliable signals reflected in above-chance ISC, even if they do not correlate with surprise as in the case of the MTG.

To assess the relative contribution of surprise to this ISFC matrix, the reviewer suggested to regress out surprise from the signal before calculating ISFC again. This would be effective if there existed a linear relationship between surprise and univariate activity, which may reflect in ISFC. However, because we did not find evidence for such an association of surprise with the BOLD signal (Supplementary Figure 4), we already know that surprise is not reflected in a univariate activation or deactivation during the surprising event time-window. Rather, surprise triggers a more complex pattern of fluctuation in activity, which is similar across participants and across regions, as reflected in ISC/ISFC. Since this fluctuation takes on a different shape for each surprising event, we find no consistent univariate pattern of response across surprise peaks (Supplementary Figure 4C). Thus, we could not regress out surprise from the BOLD based on a consistent response pattern, and could only regress out surprise as convolved with a simple HRF. This resulted in an almost identical ISFC matrix as the one presented above, confirming that the contribution of surprise to ISFC cannot be projected out as a simple univariate estimate.

Given the above, it is more informative to estimate the contribution of surprise directly on the dynamic ISFC time-course, such as in a linear regression with surprise ratings as the predictor of dynamic ISFC, which produces the same correlation values presented in Figure 2. The explained variance is given by the R squared, i.e. the squares of the correlations between surprise ratings and ISFC time-courses, now presented in new Supplementary Figure

7B (below). This is the most informative way we can present the relative contribution of surprise to DMN dynamics within our data. We see that the highest variance explained by surprise (~27%) is in the ISC of the right AG, whereas surprise hardly explains any of the variance in ISC of the left MTG.

Finally, we asked whether low correlation with surprise, such as in the ISC of the MTG, stems from low reliability of the signal, or may reflect in high ISC/ISFC values that are not exclusive to surprising events. To this end, we calculated the ISFC matrix during the 5 surprise peaks relative to baseline, presented below (Panel A) side-by-side with the matrix of the explained variance (Panel B).

Supplementary Figure 7. Inter-subject functional correlation (ISFC) at peak surprise and surprise-explained variance in the default mode network (DMN) for Sherlock. (A) Peak-SFPA for each DMN region-pair, as the ISFC values (mean of 35 fMRI participants) at event onset, averaged across the 5 peak surprising events. Black outlines denote above-chance SFPA ($p < 0.05$) via permutation testing (1000 iterations); (B) The percentage of variance explained by surprise ratings in ISFC fluctuation, for each DMN region pair. Explained variance is calculated as the squared Pearson correlation between surprise ratings (mean of 45 behavioral participants) and ISFC (mean of 35 fMRI participants) across the 49 movie events, plotted as percentage ($R^2 \times 100$).

The new peak analysis is now described in Methods, lines 420-422 and 454-461:

...In addition, we measured peak-SFPA separately for each DMN region pair, following the same steps as described earlier, but without averaging ISFC across network regions.

... significance of peak-SFPA for each region pair was determined in a permutation test, whereby, in each iteration, we randomly picked 5 events (to match the 5 peak-state events) and calculated their mean ISFC, assigning the resulting value to a null distribution separately for each pair of DMN regions. This was repeated 1000 times, resulting in a distribution of 1000 data points per region pair, representing the null hypothesis, i.e. chance-level ISFC across any 5 events. The p value for each region pair was determined by the percentile of its peak-SFPA in respect to the null distribution. Significance was determined at $p < 0.05$ (1-tail), following FDR correction across all region pairs.

Together, the two matrices clarify the relationship between ISC/ISFC and surprise within each region/region-pair. Referring directly to the reviewer's examples, though MTG exhibits higher ISC than most other DMN regions during peak-surprise events, it is not significantly selective for peak surprise (Panel A). Similarly, the low variance explained by surprise in ISC of the MTG (Panel B), suggests that its activity is not mediated by surprise, despite its high reliability throughout the movie (Supplementary Figure 6 above). In the case of the AG and MFG, surprise explains at least a quarter of their measured coactivation, and indeed, their ISC/ISFC values are significantly higher during surprising events. We can thus safely conclude that a lack of correlation with surprise does not suggest low ISC/ISFC, and that in regions most highly associated with surprise, there is still a large portion of activity to be accounted for by other processes.

These new results are now reported in the Results section, lines 160-168, where we emphasize the qualitative distinction between ISFC and SFPA:

In addition, we distinguish coactivation magnitude, as ISFC, from its association with behavior, as SFPA. Particularly, low SFPA does not suggest low coactivation in general, as demonstrated by above-chance ISFC (Supplementary Figure 6) even in region pairs showing below-chance correlation between surprise and ISFC (Figure 2). This distinction is particularly evident in the left and right middle temporal gyrus (MTG), showing high ISFC among themselves during peak surprise, though not significantly selective for surprising events (Supplementary Figure 7A). In fact, surprise explains nearly no variance in these connections, whereas it explains up to ~27% of the variance in ISFC fluctuation of other nodes such as the right angular gyrus (AG) (Supplementary Figure 7B).

2) The paper indicates that not all DMN nodes contribute equally to the propagation of surprise ratings across regions. For example, AG and PCC seem more central than other nodes. Recently few studies subdivided the DMN into subnetworks (e.g. see recent bio archive paper by Barnett and Ranganath). Does the propagation of surprise ratings match one of these subdivisions better?

It is true that not all DMN nodes are similarly affected by surprise. However, even considering only the most prominent connections associated with surprise, they cannot be assigned to a single sub-network of Barnett and Ranganath's division, nor do they fit other proposed DMN sub-divisions to the best of our knowledge. In fact, all DMN regions are engaged in surprise-dependent coactivations (even MTG, though not in its ISC, nevertheless in its ISFC with other regions), showing that surprise affects the whole network, not one sub-division exclusively. We now explicitly address this in the Results, lines 169-176:

Despite variability across region pairs in explained variance (Supplementary Figure 7B), note that all DMN regions were nevertheless engaged in surprise-dependent coactivations. Specifically, while AG, MTG (in its subcortical connections) and the middle frontal gyrus (MFG) are the most prominent nodes associated with surprise, also their coactivations with the medial prefrontal cortex (mPFC) and posterior cingulate cortex (PCC) are above chance (Figure 2). Selective coactivation of the DMN during peak surprise is also a network-wide effect, preserved after averaging across all network regions (Figure 3), as well as throughout 49 out of 55 region pairs when examined separately (Supplementary Figure 7A).

Minor comments:

1) I was worried that using a narrow window of 15TRs will make the ISFC analysis unstable. Especially in Figure 3. Would it be possible to add a supplementary analysis with window sizes of 30TRs and 60TRs?

Because our behavioral events are separated by ~30 seconds in the main analysis (Sherlock), corresponding in length to 20 TR, overlap between events would begin at ~20TR sized windows and increasingly smear the data as we go up. Nevertheless, to demonstrate the stability of ISFC, we have added a new Supplementary Figure 9, presenting the peak analysis of DMN as in Figure 3A, using ISFC time-windows of 10, 20, 25 and 30 TRs. These confirm that the increase in ISFC during surprise is significant and stable throughout window sizes 15-25 TR, though at 25 smearing is already visible. At the narrow window of 10 TRs, the analysis is indeed noisier, showing an apparent increase following peak surprise but not significantly above noise level. At the wide window of 30 TRs there is already considerable overlap between behavioral events, thus flattening the peak analysis.

Supplementary Figure 9. DMN peak analysis for Sherlock using different time-window sizes. ISFC was calculated across 10, 20, 25, or 30 TR windows. Peak SFPA – ISFC mean of 35 fMRI participants and of all network regions was averaged across the 5 peak events on each behavioral measure (e.g. ISFC during 5 most surprising events). This resulted in a mean ISFC value per network per peak-state, presented here. DMN regions were selectively coactivated during peak surprise, compared to all other peak states, as revealed by random permutation testing (1000 iterations) at $p < 0.05$. Network ISFC is plotted as mean \pm SEM across subjects, corresponding to event onset (time 0), and to each of the 14 TRs before and after the event, during each peak state (irrespective of window size for ISFC calculation, described above).

2) There appear to be some similarities in the connections between hippocampus and DMN during the ratings of surprise and during the ratings of importance and emotional intensity. Did you control for possible partial correlations across the behavioral ratings?

To the naked eye, the ISFC between DMN and HC seem to show a similar pattern of correlations with surprise, importance and intensity, though only surprise is above chance. Given collinearities among the behavioral measures themselves, we cannot rule out a relationship between DMN-HC coactivation and importance or intensity. However, by controlling for partial correlations as the reviewer proposed, we can show that they do not explain any additional variance beyond that which they share with surprise, and that surprise does, in fact, explain additional variance beyond what is shared with intensity and importance.

To examine this, we performed the same correlation analysis presented in Figure 4 for DMN and subcortical regions, but adjusted for partial correlations across all behavioral measures. Black outlines mark significance, determined in a permutation test in the same way as in the original correlation analyses. Results, presented in the new Supplementary Figure 2 below, suggest that beyond the variance explained by other behavioral measures, surprise is significantly associated with ISFC of the DMN and its coactivation with HC and the striatum. By contrast, we see weak correlations with emotional intensity and with importance while controlling for surprise and all other measures.

To validate the stability of this result, we performed two additional control analyses. First, we used a multiple regression GLM to test the linear relationship between each behavioral predictor and the response time-course, while keeping the remaining predictors constant. Second, we trained a Lasso-regularized regression using cross validation (10 folds of ~5 events each), and examined betas given the minimum MSE. Both of these yielded similar results as the adjusted correlation analysis, confirming a unique correlation of surprise with DMN-HC coactivation.

Supplementary Figure 2. Correlation analysis adjusted for partial correlations among behavioral measures in Sherlock. To assess the contributions of collinearities among behavioral measures to SFPA results, we performed correlation SFPA as in the original analysis, but adjusted for partial correlations between each behavioral measure and all others. Measures of highest behavioral collinearity with surprise were emotional intensity and importance. Results show that after adjusting for partial correlations among all behavioral measures, surprise is significantly associated with ISFC of the DMN and subcortical regions, whereas emotional intensity and importance show no above-chance SFPA. Pearson correlations were calculated between each behavioral time-course (mean of 45 behavioral participants), adjusted for partial correlations with all other measures, and ISFC of each region-pair (mean of 35 fMRI participants), across the time-course of movie events. Black outlines denote above-chance correlations at $p < 0.05$ (corrected), determined by random permutation testing (1000 iterations).

3) I was confused by the wording you used to describe Fig 3C: “repetitions of a similar event within the same scene”. I assume the same event was repeated across different scenes, and not that the same event was repeated three times within the same scene. If it were repeated within the same scene how could you detect the difference between the three repetitions within the same time window?

We apologize for the confusion. By “scene” we mean the director-cut definition of a scene, which can be minutes long. By “event” we refer only to a single moment within the scene. We now describe this in more detail in Results, lines 106-114:

... we see an example for this in the movie, in a scene depicting a press conference, in which journalists ask police detectives about a series of alleged suicides. At 3 different times within this press-conference scene, a similar momentary event occurs. The first time, peak surprise is triggered when all journalists at once receive a text message saying “wrong”. Later during the scene, the same mass-text event repeats twice more. Each of the 3 mass-text events corresponded to a separate ISFC window, with no overlap between them, thus we can examine them separately. As demonstrated in Figure 3C, mean ISFC of the DMN plummeted during the second occurrence of this event, and remained low during the third, whereas DAN and Vis exhibited different response patterns.

REVIEWERS' COMMENTS:

Reviewer #1 (Remarks to the Author):

The authors have done a superb job at addressing the previous set of recommendations.

Reviewer #2 (Remarks to the Author):

I applaud the authors for the detailed revision, which addresses all of my concerns. This is a fantastic paper, which will provide an important contribution to the field.